# Postprandial Glucose Response after Consuming Low-Carbohydrate, Low-Calorie Rice Cooked in a Carbohydrate-Reducing Rice Cooker

**DOI:** 10.3390/foods11071050

**Published:** 2022-04-06

**Authors:** Hyejin Ahn, Miran Lee, Hyeri Shin, Heajung-angie Chung, Yoo-kyoung Park

**Affiliations:** 1Department of Gerontology (AgeTech-Service Convergence Major), Kyung Hee University, Yongin 17104, Korea; hjahn@khu.ac.kr (H.A.); zisoa@khu.ac.kr (H.S.); 2Department of Medical Nutrition, Kyung Hee University, Yongin 17104, Korea; conan_6612@naver.com; 3Department of Korean Cuisine, Jeonju University, Jeonju 55069, Korea; 4Department of Medical Nutrition (AgeTech-Service Convergence Major), Kyung Hee University, Yongin 17104, Korea; 5Department of Food Innovation and Health, Graduate School of East-West Medical Science, Kyung Hee University, Yongin 17104, Korea

**Keywords:** low-carb rice, carbohydrate-reducing rice cooker, blood glucose, sensory evaluation, visual analog scale

## Abstract

This study evaluates whether blood glucose response differs upon consuming rice cooked in a carbohydrate (carb)-reducing rice cooker. Rice cooked this way exhibited 19% reduced total carbohydrate (34.0 ± 0.3 vs. 27.6 ± 0.9 g/100 g rice) and 20% reduced total calorie (149.0 ± 1.0 vs. 120.8 ± 3.7 kcal/100 g rice) contents. We measured the blood glucose response (at 0, 15, 30, 45, 60, 90, and 120 min) in 13 healthy participants after consuming 6 different rice types: regular white rice (regular WR, 50 g of available carbohydrate (AC)), low-carb WR with equivalent weight as regular WR (low-carb WR (EW)), low-carb WR with equivalent carb as regular WR (low-carb WR (EC), regular mixed-grain rice (regular MR), low-carb MR (EW) as regular MR, and low-carb MR (EC) as regular MR. All rice samples were prepared in an electric carb-reducing rice cooker. Postprandial blood glucose, sensory, and appetite were assessed after each test meal. The incremental area under the curve of 15 and 30 min after rice consumption was significantly lower in low-carb WR (EW) than that in regular WR. These results suggest possible health benefits of low-carb WR in reducing early postprandial spikes in blood glucose level without significant differences in satiety and satisfaction.

## 1. Introduction

Rice is globally among the most popular food grains and is currently consumed by more than half of the global population [1,2]. White rice (WR), primarily consisting of starch, is produced through a series of mechanized processes, including hulling and milling, and it is globally the predominant rice type consumed [3,4]. The average glycemic index (GI) of WR is higher than that of whole grains [5,6]. In addition, WR is the primary contributor to dietary glycemic load (GL) in populations consuming rice as a staple food [6]. WR has been turned into an unhealthy staple crop by the conclusions of repeated studies reporting that it could lead to blood glucose increase and the development of type 2 diabetes mellitus (T2D) [7,8]. In large-scale human observational studies among various populations and several cohort studies, high GI food intake and high GL diets were associated with an increased risk of developing T2D [9,10].

Reducing carbohydrate (carb) intake is attracting growing attention from people who want to eat healthily [11]. Raising awareness of healthy eating and increasing the popularity of low-carb products in the growing global population could result in an expected market growth rate of 6.40% in the forecast period of 2020–2027 [12]. Reducing carb intake helps in managing insulin resistance, glycemia, and diabetes in patients with diabetes [13,14]. Related beneficial effects on body weight, cardiovascular risk indices, and other health indicators were also reported, even in people without diabetes [15,16]. WR is an important staple food in multiple Asian countries such as China, India, Japan, and Korea [17]. If it is not possible to suggest a WR substitute, so it is necessary to suggest a healthier way to consume WR. Recent efforts aim to enable the healthy consumption of WR by changing its cooking method.

Carb-reducing electric rice cookers theoretically reduce starch contents in rice. Recently, with the increasing number of patients with diabetes and people who want to consume healthy meals, the carb-reducing rice cooker is a promising device in the kitchen appliance market. In particular, interest in carb-reducing rice cookers is growing in Asia, where rice is the staple food. Various carb-reducing rice cookers are manufactured, and the effect of reducing starch varies from a minimum of 20% to a maximum of 50%. 

The basic principle of the carb-reducing rice cooker is to boil rice in excess water (3–4 times the weight of the rice), and starch contents from the rice are thus dissolved in the water and removed. Starch dissolved in rice is removed, resulting in low-carb rice. Low-carb rice cookers are divided into two types, the siphon and wash-down types, according to the method of draining the excess water (Figure 1). The siphon-type carb-reducing rice cooker is characterized by a tray in the inner pot for cooking. As the inside of the rice cooker heats up rapidly, a pressure difference occurs in the inner pot. The pressure difference pushes the water upward, and the water collects in the tray. After the excess water with the dissolved rice starch is removed on the tray, the rice is cooked the same way as in a general rice cooker. The wash-down type carb-reducing rice cooker is characterized by a basket and an inner pot that is deeper than the basket. Rice is put inside the basket, and the bottom of the inner pot is filled with water that does not submerge the basket. The rice is cooked by placing a basket of it in the inner pot filled with water. As the inner pot is heated, boiling water repeatedly splashes into the basket, and the rice starch dissolves into the water. However, there are some complaints that the wash-down type rice contains too much water. 

These carb reduction effects claimed by companies are based on food composition analysis, and no clinical research data are available on the effect of carb-reducing rice consumption on blood glucose response.

The glucose response after consuming carbohydrate-containing food relative to a carbohydrate reference food, typically white rice or write bread, is normally measured using the glucose index (GI). The primary purpose of this study was to determine whether low-carb rice cooked in a siphon-type carb-reducing rice cooker could reduce blood glucose response in healthy adults. The secondary purpose was to perform sensory and appetite evaluation after low-carb rice consumption.

## 2. Materials and Methods

### 2.1. Study Design

This study was conducted to evaluate how consuming low-carb rice cooked in a carb-reducing rice cooker affects blood glucose response and appetite, and to perform sensory evaluation. The study was conducted in accordance with the Declaration of Helsinki, and the protocol was approved by the Ethics Committee of the Institutional Review Board (IRBs) of Kyung Hee University in September 2021 (no. KHGIRB-21-420).

### 2.2. Study Participants

We recruited 13 healthy participants (4 men and 9 women) from the campus and the community. The inclusion criterion was being at least 35 years old. Exclusion criteria were as follows: patients with uncontrolled diabetes abd people unable to perform a finger-prick blood test due to psychological fear. All subjects gave their informed consent to participate in the study before starting.

### 2.3. Test Rice Preparation

Rice samples were cooked in a carb-reducing electric rice cooker (CALOFREE CFRC-IH600, CADEAUS Co., Ltd., Seoul, Korea). The carb-reducing rice cooker used in the study was a siphon type having the same composition as that of an electric rice cooker with only the inclusion of a tray (Figure 2). When cooking in regular rice mode, the tray was not placed on the inner pot for cooking; this was only when using the carb-reducing rice mode. This siphon-type rice cooker is an induction heating (IH) nonpressure rice cooker with the function of a regular rice cooker. It was designed as a nonpressure IH system because boiling water must freely splash up and down inside the rice cooker to cook low-carb rice. The IH system quickly heats the inner pot to high heat. This boils the water quickly, allowing for the starch to dissolve in the water before the rice is overcooked, and creates a high pressure difference in the inner pot, effectively pushing excess water up through the tray. After the rice is cooked, the water accumulated in the tray should be removed.

Rice (white rice (WR): polished rice 300 g; mixed-grain rice (MR): 210 g polished rice + 90 g mixed-grain rice) was washed three times with sufficient distilled water [18,19,20]. Low-carb rice samples were cooked with a water/rice ratio (W/R) of 4.0 (1200 mL) without immersing in water in the low-carb rice mode of the rice cooker for 35–45 min. General rice samples were cooked with a W/R ratio of 1.2 (300 mL) after soaking the rice in water for 30 min in the general rice mode of the rice cooker for 45–50 min. The multigrain used in the study was a commercially available 11 Mixed Grains (Daesang Agricultural Farming Corp. Co., LTD., Wanju, Korea), which is composed of glutinous, brown, glutinous brown, black glutinous brown, red–brown, green–brown, red–brown, compressed, and sticky barley rice, and two type of millet rice. After the rice had been automatically cooked, it was left on the warm setting for an additional 8 min and mixed well. Figure 3 presents the weight change diagram of rice and water before and after cooking. The test rice was cooked in the Nutrition Living Lab at Kyung Hee University.

### 2.4. Test Rice Nutrient Analysis

The cooked rice was gently fluffed with a spatula and transferred into a container with a plastic lid for nutrient analysis. Freshly cooked rice was sent to Société Générale de Surveillance (SGS, Seoul, Korea) for nutrient analysis.

### 2.5. Study Procedure

Subjects were provided with a verbal explanation of the study and directed to follow the instructions on the study description document. Participants were instructed to fast overnight for 12 h before their participation in the study. In addition, subjects were encouraged to eat their usual meals and conduct their daily activities the day before their participation. The participants visited the living lab 7 times, consumed 6 types of test foods, and performed a finger-prick blood glucose test, visual analog scale (VAS) test for the assessment of hunger, feeling of fullness, and desire to eat, and sensory evaluation. 

We followed the official method for determining the GI in foods instructed by the Organization for International Standardization (ISO) [21,22] Participants were tested for oral glucose tolerance at the first visit, followed by six visits to participate in each food testing session. All tests were conducted at approximately 09:00 a.m.–12:00 p.m. Blood glucose levels were tested on an empty stomach (0 min) and at 15, 30, 45, 60, 90, and 120 min after test food consumption. Only individuals with fasting glucose levels below 110 mg/dL could participate in the study. 

The six test rice types were as follows: (1) regular WR (147 g one small bowl, 50 g of available carbohydrate (AC)), (2) low-carb WR with equivalent weight (EW) as regular WR (147 g serving, 40 g AC), (3) low-carb WR with equivalent carb (EC) as regular WR (181 g serving, 50 g AC), (4) regular MR (144 g one small bowl, 50 g AC), (5) low-carb MR (EW) (144 g serving, 40 g AC), and (6) low-carb MR (EC) (181 g serving, 50 g AC).

The blood glucose levels of participants were measured with the finger-prick method using an Accu-Chek Performa Blood Glucose Meter (Accu-Chek, USA). Each blood glucose level after the test meals was used to calculate the incremental area under the curve (iAUC) [23].

### 2.6. Sensory Evaluation

Participants rated the six types of cooked rice for intensities of individual attributes, including appearance, flavor, taste, texture properties, and overall preference, on a 9-point structured scale (1 = extremely low intensity and 9 = extremely high intensity) [24]. Cooked rice samples of approximately 15 g were prepared in plastic containers and tested by the panelists. Samples were coded with random three-digit numbers. Participants were presented with six samples in random order on a tray with a glass of drinking water at each testing session. Samples were served warm, and participants were instructed to cleanse their palates with water prior to testing a new sample. Glossiness, color, and intactness were first evaluated as appearance characteristics of the cooked rice. The flavor was roasted and nutty, and it smelled like deliciously cooked rice, but smelled undercooked before putting it into the mouth. The test foods were evaluated for sweetness, roasted nutty flavor, and overall taste. During the few first bites, springiness, chewiness, and moistness were assessed by compressing the sample between the molars and evaluating the force required to bite through the sample. The adhesiveness and cohesiveness of the cooked rice mass were evaluated during the chewing stage as the degree to which the sample adhered to the teeth or palate. 

### 2.7. Hunger and Appetite Evaluation

For the assessment of hunger, desire to eat (DTE), and feeling of fullness, we used validated VAS [25]. Participants were asked to rate their levels of hunger (“How hungry do you feel at this moment?”), DTE (“How strong is your desire to eat at this moment?”), and feeling of fullness (“How full does your stomach feel at this moment?”), using these scales presented one at a time. Subjects marked their rating on a 100 mm line anchored at 0 (“I am not hungry/satiated at all”; “I have no desire to eat at all”) and 100 (“I have never been hungry/satiated”; “I have never had more desire to eat”). These evaluations were performed by participants before consuming the test meals (basal time, 0 min) and were repeated after the end of the meal at 60 min intervals over a total period of 180 min (0, 60, 120, and 180 min).

### 2.8. Anthropometric Measures

Height, body mass index, and body composition were measured using a stadiometer (BSM330, Inbody, Seoul, Korea) and bioimpedance (Inbody770, Seoul, Korea). Participants stood barefoot with minimal clothes on in a straight position, and palms were facing the thighs, so that the posture was clear.

### 2.9. Sample Size and Statistical Analysis

Sample size was determined on the basis of the method of the International Organization for Standardization [18]. This suggests that using 10 subjects to measure glycemic response to a single food could exhibit adequate power and accuracy.

All data are expressed as mean ± standard deviation (SD) values. The iAUC and blood glucose responses were analyzed at each test time using an independent Student’s t-test. Statistical analysis was performed using the IBM SPSS 25 statistical program (IBM, New York, NY, USA). Statistical significance was set at *p*-values of *p* < 0.05.

## 3. Results

### 3.1. Participants

Of the 17 screened participants, 13 eligible (4 males and 9 females) were selected. Table 1 shows the baseline characteristics of the participants who completed the study.

### 3.2. Test Food Nutrient Contents

Table 2 presents the nutrient contents of the test meals. Compared with regular rice, low-carb rice had less carbohydrate (regular WR: 34.0 ± 0.3; regular MR: 34.8 ± 0.4 vs. low-carb WR: 27.6 ± 0.9; low-carb MR: 28.0 ± 1.4) and less energy (regular WR: 149.0 ± 1.0; regular MR: 154.4 ± 2.3 vs. low-carb WR: 120.8 ± 3.7; low-carb MR: 129.3 ± 2.9). The moisture content of low-carb rice was higher than that of regular rice (regular WR: 62.9 ± 0.3; regular MR: 61.6 ± 0.4 vs. low-carb WR: 70.1 ± 0.8; low-carb MR: 68.0 ± 0.6).

### 3.3. Incremental Area under Glucose Curve (iAUC Blood Glucose) of Test Foods

Figure 4 shows the *i*AUC blood glucose of the test foods. Blood glucose *i*AUC after low-carb WR (EW) intake was significantly lower at the time points of 0–15 and 0–30 min compared to regular WR (0–15 min: *p* < 0.05; 0–30 min: *p* < 0.05). However, no significant difference could be observed in *i*AUC blood glucose between regular MR and low-carb MR.

### 3.4. Postprandial Blood Glucose Changes after Test Food Consumption

Table 3 shows postprandial blood glucose changes after test food consumption. Compared with regular rice, average blood glucose levels at each time after eating low-carb rice were low. However, no significant difference could be observed in blood glucose levels among test foods.

### 3.5. Sensory Evaluation

Sensory profile analysis could objectively evaluate the quality of the low-carb rice by quantifying appearances, flavor, taste, texture properties, and overall preference. Table 4 displays the sensory evaluation results.

The sensory evaluation of low-carb rice cooked in a carb-reducing rice cooker showed no significant difference with regular rice in appearance (glossiness, color, and intactness), flavor (roasted nutty flavor, flavor of cooked rice, and smell of undercooked rice), taste (sweet taste, roasted nutty flavor, and overall taste), texture properties (springiness, chewiness, adhesiveness, cohesiveness, and moistness), and overall preference.

### 3.6. Hunger and Appetite Evaluation

Change in hunger, feeling of fullness, and desire to eat after low-carb rice intake were evaluated using the VAS (Figure 5). The VAS test showed no significant difference in hunger, feeling of fullness, and desire to eat for 3 h after consuming of low-carb rice compared to regular rice.

## 4. Discussion

This study was conducted to evaluate whether consuming low-carb rice cooked in a carb-reducing rice cooker affects blood glucose response. In addition, we investigated the difference in sensory, hunger, fullness characteristics, and the DTE between low-carb and regular rice. Our results showed that the iAUC of blood glucose was significantly reduced at 15 and 30 min after low-carb WR intake compared to regular WR. Sensory characteristics, and hunger, fullness, and DTE 3 h after eating values showed no difference between low-carb rice and regular rice.

Using a carb-reducing rice cooker produced ~20% fewer calories and carbohydrate contents than those of the regular rice cooking method. Moisture content is a determinant of the taste of rice, and the optimal amount of water for delicious rice is 1.2 to 1.3 times [26]. Several carb-reducing rice cookers on the market promote various starch-reducing effects, such as 20–40%, but the cooking method for regular and low-carb rice is not standardized and has not yet been compared. In this study, regular rice was cooked in water that was 1.2 times the weight of the rice according to the recipe for rice consumed in daily life, while low-carb rice was cooked in water at 4 times the weight of rice. This method produced the best taste and was used for the experiments. 

Low-carbohydrate diets, which have recently attracted much attention, are effective in weight loss and the prevention of chronic diseases such as diabetes by regulating the glucose–insulin axis [27]. In women with gestational diabetes, or overweight or obese adults, low-carbohydrate mixed meals had significantly reduced iAUC in blood glucose and/or insulin [27,28]. In addition, the intake of a low-carbohydrate ketogenic diet for 12 weeks had an effect in reducing body weight and BMI [29]. Gene expression involved in nutrient metabolism is also different during day and night and depends on meal composition [30]. Therefore, manipulating the meal type at night (e.g., more protein and less carbohydrate) may mitigate the regulation of glucose uptake, and fatty acid synthesis and oxidation [30]. Previous studies mainly evaluated the effect of a low-carbohydrate diet with increased protein and fat content. This study evaluated the blood glucose response in low-carb rice intake. The effect of reducing the blood glucose levels of low-carb rice suggests that the blood glucose can be controlled, not through a high-protein or -fat meal, and through daily meals.

WR is among the most important staple foods for the global population [1,2]. Asian people consume cooked rice with almost every meal. The consumption of WR mainly composed of starch can be a burden for people with diabetes and impaired glucose tolerance who need to control their blood glucose [5,6]. Reducing the burden of blood glucose increase in the intake of WR (high-GI and frequently consumed food) could reduce the stress of diet control in people who need to control their blood glucose, and suggests the possibility of blood glucose control through daily meal intake [7,8]. From the results of the study, iAUC at 15 and 30 min after the consumption of low-carb WR was significantly lower than that of regular WR. Delaying the increase in blood glucose is beneficial for blood glucose management and can help in lowering postprandial blood glucose spikes, resulting in stable blood glucose levels [31]. Therefore, these results show the positive effects of low-carb WR against postprandial spikes in blood glucose, and suggest options for those who hesitate to consume white rice due to the burden of blood glucose control.

The postprandial glucose response to carbohydrate meals is not only determined by the amount of available carbohydrate, but also by fat, protein, and various nutrients [32,33]. Soluble dietary fiber in foods also influences the glycemic response after a meal [32,33]. The consumption of intact grains containing at least 4 g of β-glucan and 30–80 g of available carbohydrate is reportedly required to reduce postprandial blood glucose [34]. In addition, studies reporting on the postprandial blood glucose reduction effect of ingesting barley and brown rice also confirmed the effect of ingestion of 100% barley and 100% brown rice, and not mixed grains with WR [35,36]. A previous study reported that MR with a ratio of less than 30% is the most preferrable [37]. In another study, rice with a ratio of mixed grains was 40% higher than rice with a ratio of 50% [38]. Therefore, in this study, the ratio of mixed grains was selected as 40% for mixed rice that people can eat deliciously by reflecting their preferences. The mixed grains used in this study were a mixture of glutinous, brown, glutinous brown, black glutinous brown, red–brown, green–brown, red–brown, compressed, sticky barley, and millet rice. The MR provided to the subjects in the study was cooked by mixing 40% mixed grains and 60% WR, a commonly consumed mixed rice recipe in Korea. Our results showed that the intake of MR with a mixed-grain ratio of 40% did not affect postprandial blood glucose levels. We assumed that, since there was only 60% of WR in the MR, further reduction in blood glucose of the low-carb MR could not be not easily attained. 

To comply with diabetic diets, a major determinant of the effect of the diet on blood glucose management, foods that are not palliative hinder sustaining it for a long period. Low-carb rice in this study had about 20% fewer calories and less carbohydrate than regular rice did (regular WR: 140.9 kcal; low-carb WR: 120.8 kcal; regular MR: 154.4 kcal; low-carb MR: 129.3 kcal), but there was no noticeable difference in the sensory evaluation (appearance, flavor, taste, texture properties, preference) and the feeling of hunger, fullness, and DTE in consumption. When people are on a low-carbohydrate diet and consume less rice, fullness is normally low, and hunger and DTE are high, and the diet is usually accompanied by additional food intake, which leaves a low-carb diet with no health benefits. However, results of our study show that maintaining a low-carb diet with a carb-reducing rice cooker seems feasible because eating low-carb rice did not affect hunger or appetite.

Visual cues for food can affect food intake [39,40]. In this study, when low-carb rice of equivalent weight (low-carb WR (EW); low-carb MR (EW), and low-carb rice) with the same carbohydrate amount (low-carb WR (EC); low-carb MR (EC) was served with regular rice (regular WR; regular MR), even when their weight differed by up to 30 g (about 1–1.5 spoon), the difference was not noticeable with the naked eye. This may have affected the feeling of satiety, as subjects visually perceived a similar amount of rice. When regular WR (147 g, 50.0 g carbohydrate, 219.0 kcal) and low-carb WR (EW) (147 g, 40.6 g carbohydrate, 177.6 kcal) were consumed, there was no difference in the feeling of hunger, fullness, and urge to eat for 3 h after eating. These results suggest that consuming low-carb rice (eventually lower calories with no significant empty feeling) can be used as dietary therapy for body-weight loss.

The flavor of low-carb rice is closely related to the sustainability of low-carb rice consumption in daily life. Even if there are foods that help in controlling blood glucose due to their low-carb content, it is difficult to maintain intake if they do not taste good. Therefore, low-carb rice does not exhibit sensory difference, such as regarding appearance, flavor, taste, texture properties, and preference, compared to regular rice. 

Common cooking principles of the carb-reducing rice cooker are as follows. First, put rice and excess water (about four times or more the weight of rice) in a rice cooker and boil them together. Next, after the starch in the rice had dissolved in the water, leaving only an appropriate amount of water, rice is cooked after removing the extra water. When the carb-reducing rice cooker was first released, it was mainly a wash-down type carb-reducing rice cooker that drained water to the bottom. In this rice cooker, soft- or liquid-type rice was prepared by exposing the rice to the water at the bottom. The siphon-type carb-reducing rice cooker, which was released later, removes rice water with the upper tray and cooks the rice. However, because of the cooking principle for reducing starch in the carb-reducing rice cooker, starch in the rice is removed using water, and low-carb rice can be cooked with a lot of water. 

Some limitations of this study should be noted. First, no difference in blood glucose response was found between regular and low-carb mixed rice. Second, the blood glucose response test was not performed 180 min after eating to see how long the effect can last. Third, the study was performed using the finger-prick blood test for convenience, but we thus could not assess insulin hormone concentrations for more accurately assessing the glycemic response.

The strengths of the study are as follows. First, this is the first study to evaluate and publish the blood glucose response of low-carb rice cooked with a carb-reducing rice cooker. Second, we found a way to control blood glucose with low-carb rice using WR, the staple food of Asian countries. Third, there was no difference in the feeling of fullness, hunger, and DTE between the consumption of low-calorie low-carb WR and regular rice.

## 5. Conclusions

The study is the first clinical trial to evaluate the glycemic response effect of low-carb rice cooked in a carb-reducing rice cooker. Low-carb and regular rice used in the study was cooked with a recipe that could produce the optimal flavor, and the low-carb rice had about 20% fewer calories and less carbohydrate than those of regular rice. Low-carb rice had no difference in sensory characteristics, fullness, hunger, and DTE compared to regular rice. These results indicate the health benefit of low-carb WR on postprandial spikes in blood glucose levels. The possibility of low-carb rice as a substitute for regular rice can be carefully claimed. Eating less carbohydrate (20% less carbohydrate and kcal per each meal can add up to 35 g carbohydrate and 150 kcal deficit per day) without feeling deprived contributes to improving the quality of life of people who need blood glucose control, such as patients with diabetes or impaired glucose tolerance.

## Figures and Tables

**Figure 1 foods-11-01050-f001:**
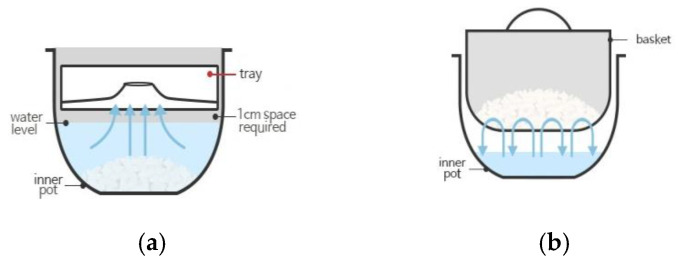
Schematic design of siphon and wash-down carb-reducing rice cooker types. (**a**) Siphon-type carb-reducing rice cooker characterized by a tray in the inner pot. As the inside of the rice cooker heats up rapidly, pressure difference occurs in the inner pot. (**b**) Wash-down type carb-reducing rice cooker displays a basket and an inner pot that is deeper than the basket.

**Figure 2 foods-11-01050-f002:**
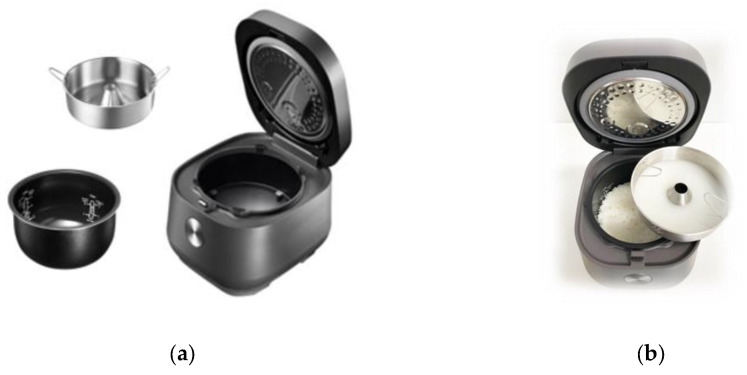
Composition of a siphon-type carb-reducing rice cooker. (**a**) Carb-reducing rice cooker component; (**b**) Appearance after cooking low-carb rice.

**Figure 3 foods-11-01050-f003:**
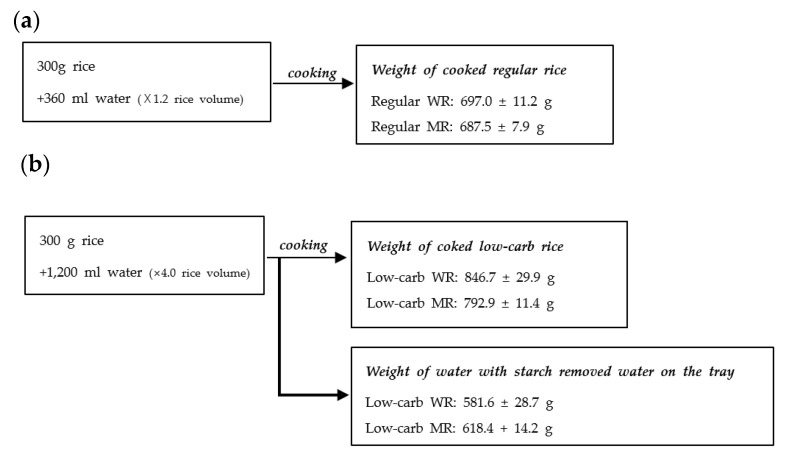
Weight change diagram of rice and water before and after cooking: (**a**) regular rice; (**b**) low-carb rice. Low-carb rice samples were cooked with water/rice (W/R) ratios of 4.0 (1200 mL) without immersion in water in low-carb rice mode of the rice cooker for 35–45 min. General rice samples were cooked with W/R ratios of 1.2 (300 mL) after soaking the rice in water for 30 min in the ‘general rice mode’ of the rice cooker for 45–50 min. White rice (WR): 300 g polished rice; mixed-grain rice (MR): 210 g polished rice + 90 g mixed-grain rice).

**Figure 4 foods-11-01050-f004:**
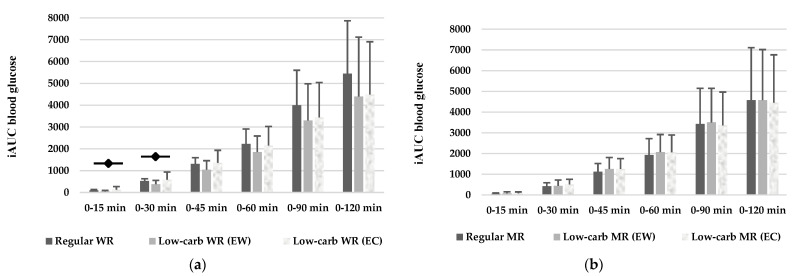
Incremental area under glucose curve of test foods Significant differences by independent t-test between regular WR *i*AUC and low-carb WR (EW) *i*AUC indicated with ◆ (*p* < 0.05). EC, equivalent carbohydrate as regular rice; EW, equivalent weight as regular rice; *i*AUC, incremental area under the curve; WR, white rice; MR, mixture of mixed grains to WR (WR/MR ratio is 7:3). (**a**) White rice; (**b**) Mixed-grain rice.

**Figure 5 foods-11-01050-f005:**
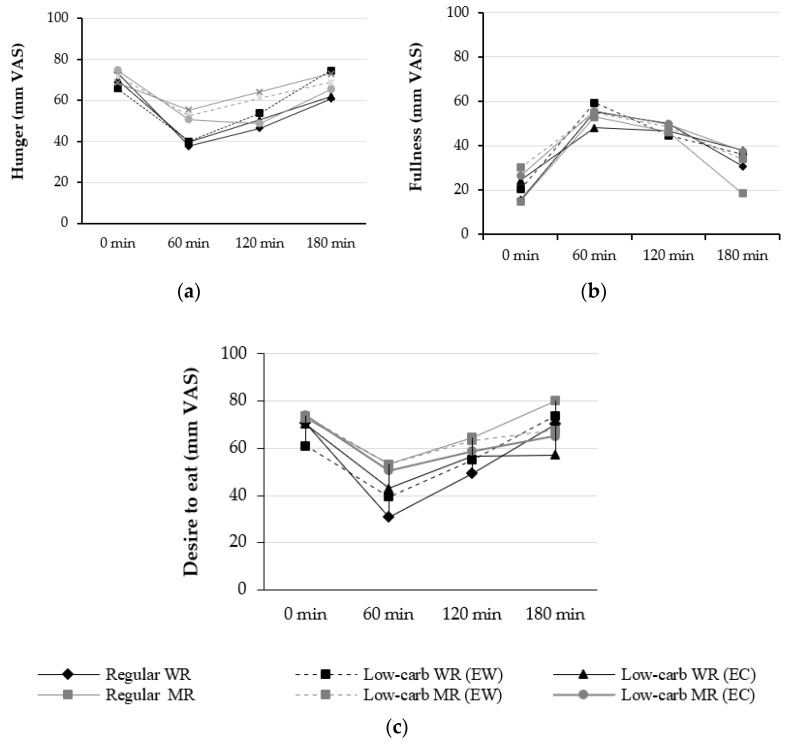
Change in hunger, fullness, and desire to eat using a visual analog scale. EC, equivalent carbohydrate as regular rice; EW, equivalent weight as regular rice; WR, white rice; VAS, visual analog scale; MR, mixture of mixed grains to WR (WR/MR ratio is 7:3). Data were not significantly different. (**a**) Hunger; (**b**) Feeling of fullness; (**c**) Desire to eat.

**Table 1 foods-11-01050-t001:** Characteristic of subjects.

Classification	Subjects (n = 13)
Age (years)	56.8 ± 12.1 ^1^
Sex (male/female)	4/9
Height (cm)	162.2 ± 10.6
Body weight (kg)	64.7 ± 14.7
Body mass index (kg/m^2^)	24.5 ± 4.4
% Body fat (%)	32.4 ± 5.3
Skeletal muscle mass (kg)	24.0 ± 6.9
Waist–hip ratio	0.90 ± 0.05
Fasting blood glucose (mg/dL, range)	99.5 ± 15.0 (82–120)

^1^ Data are mean ± standard deviation (SD).

**Table 2 foods-11-01050-t002:** Test food compositions.

Test Food	Energy (kcal/100 g)	Carbohydrate (g/100 g)	Protein (g/100 g)	Fat (g/100 g)	Ash (g/100 g)	Dietary Fiber (g/100 g)	Water (g/100 g)
Regular WR	149.0 ± 1.0 ^1^	34.0 ± 0.3	2.6 ± 0.0	0.4 ± 0.0	0.1 ± 0.0	0.4 ± 0.0	62.9 ± 0.3
Low-carb WR	120.8 ± 3.7 *	27.6 ± 0.9 *	1.7 ± 0.8	0.5 ± 0.1	0.1 ± 0.0	0.4 ± 0.0	70.1 ± 0.8 *
Regular MR	154.4 ± 2.3	34.8 ± 0.4	2.9 ± 0.1	0.5 ± 0.1	0.2 ± 0.0	0.5 ± 0.0	61.6 ± 0.4
Low-carb MR	129.3 ± 2.9 **	28.0 ± 1.4 **	3.3 ± 1.0	0.6 ± 0.2	0.2 ± 0.0	0.5 ± 0.0	68.0 ± 0.6 **

WR, white rice; MR, mixture of mixed grains to WR (WR/MR ratio is 7:3). All experiments were carried out in triplicate and repeated 3 times. ^1^ Data represent mean ± standard deviation (SD) calculated by data of cooked rice. * *p* < 0.01 compared to regular WR; ** *p* < 0.01 compared to regular MR, after independent *t*-test.

**Table 3 foods-11-01050-t003:** Postprandial blood glucose changes after the consumption of test foods.

Test Foods (Serving)	Blood Glucose at Each Time (mg/dL)						
	0 min	15 min	30 min	45 min	60 min	90 min	120 min
Glucose (50 g)	101.1 ± 15.2 ^1^	135.7 ± 26.8	164.2 ± 32.2	165.2 ± 43.9	169.8 ± 50.4	156.8 ± 61.2	130.9 ± 55.6
Regular WR	99.6 ± 13.8	112.7 ± 12.4	144.9 ± 17.6	159.0 ± 31.0	161.8 ± 41.7	157.1 ± 44.6	141.5 ± 40.1
Low-carb WR (EW)	99.7 ± 12.3	105.3 ± 11.4	139.8 ± 22.0	149.3 ± 25.3	158.8 ± 37.4	139.7 ± 46.9	133.9 ± 42.1
Low-carb WR (EC)	99.5 ± 17.1	116.5 ± 33.0	143.6 ± 24.0	158.8 ± 38.7	146.1 ± 34.6	138.6 ± 41.1	130.2 ± 40.0
Regular MR	100.5 ± 16.4	107.5 ± 18.0	141.4 ± 25.8	153.4 ± 31.8	155.4 ± 43.7	145.2 ± 38.7	132.3 ± 46.5
Low-carb MR (EW)	96.5 ± 13.9	105.8 ± 14.6	140.8 ± 23.9	154.9 ± 32.0	147.2 ± 32.1	141.2 ± 46.6	123.5 ± 35.1
Low-carb MR (EC)	100.6 ± 19.4	110.3 ± 21.1	147.7 ± 30.2	153.5 ± 39.4	149.7 ± 43.0	139.5 ± 44.4	132.5 ± 45.8

^1^ Data represent mean ± standard deviation (SD) and were not significantly different. Regular WR (147 g), low-carb WR (EW, 147 g), and low-carb WR (EC, 181 g) contained 50, 40, and 50 g carbohydrate, respectively. Carbohydrate content of regular MR (144 g), low-carb MR (EW, 144 g), and low-carb MR (EC, 179 g) was 50, 40, and 50 g, respectively. EC, equivalent carbohydrate as regular rice; EW, equivalent weight as regular rice; WR, white rice; MR, mixture of mixed grains to WR (WR/MR ratio is 7:3).

**Table 4 foods-11-01050-t004:** Sensory evaluation of test foods.

Test Foods	Appearances	Flavor	Taste	Texture Properties	Overall Preference
Regular WR	5.7 ± 1.7 ^1^	4.7 ± 2.2	5.8 ± 1.6	5.6 ± 2.0	6.7 ± 1.2
Low-carb WR	6.4 ± 1.5	5.1 ± 2.3	5.8 ± 1.5	5.4 ± 2.1	6.3 ± 1.2
Regular MR	6.1 ± 1.6	4.9 ± 1.9	5.4 ± 1.5	5.4 ± 1.9	5.9 ± 1.7
Low-carb MR	6.4 ± 1.5	5.1 ± 2.2	5.9 ± 1.6	5.5 ± 1.8	6.5 ± 1.5

^1^ Data represent mean ± standard deviation (SD), were calculated by data of cooked rice, and were not significantly different. Appearance of test foods evaluated by their glossiness, color, and intactness. Flavor investigated by examining degree of roasted nutty flavor, flavor of cooked rice, and smell of undercooked rice. Test food taste was evaluated by sweet taste, roasted nutty flavor, and overall taste. Texture properties of test foods investigated by examining degree of springiness, chewiness, adhesiveness, cohesiveness, and moistness. WR, white rice; MR, mixture of mixed grains to WR (WR/MR ratio is 7:3).

## Data Availability

Not applicable.

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
