# Peer review of "Postprandial Glucose Response after Consuming Low-Carbohydrate, Low-Calorie Rice Cooked in a Carbohydrate-Reducing Rice Cooker"

_foods, 2022, doi:10.3390/foods11071050_

Round 1
Reviewer 1 Report
I have gone through in detail the reading of this manuscript. In the course, I could find a report that "low-carb rice cookers failed to reduce the level of carbohydrate compared to traditional rice cookers". I have points below to raise about the study.
- I wonder why investigators chose only a siphon-type cooker and not the wash-down !
- The authors did not disclose the composition of mixed-grain rice.
- Rice was washed three times with water before placing in the cooker. What was the loss of macro-and micronutrients by the washing?
- Figure 3 (b): Is it the 'weight of water with the starch removed water' or the weight of rice with the starch removed water?
- Table 2. Protein content in low-carb MR appears significantly higher than others. What may be the reason? How many cooking replicates were taken to complete proximate analysis? Section 3.2 statistical analysis to find differences either within or between the groups are missing. they are important to find strength in the study and analysis.
- Table 4. after 60, 90, and 120 mins, there appears no difference in observed blood glucose level. Although, the authors have not done statistical analysis. Then where does the title matches with the study results?
- Discussion finds very poor support from relevant references and also convincing arguments.
Author Response
The revised manuscript was sent by e-mail (foods@mdpi.com).
Thank you for review's comments and opportunities to improve our manuscript ‘Postprandial glucose response after consuming low-carbohydrate, low-calorie rice cooked in a carbohydrate-reducing rice cooker’
We thank the reviewers for their detailed and specific comments for the manuscript to be much improved.
Sincerely yours,
-------------------------------------------------------------------------------
I have gone through in detail the reading of this manuscript. In the course, I could find a report that "low-carb rice cookers failed to reduce the level of carbohydrate compared to traditional rice cookers". I have points below to raise about the study.
- I wonder why investigators chose only a siphon-type cooker and not the wash-down !
→We selected a siphon-type rice cooker that can cook rice that is organoleptically similar to rice cooked with traditional rice cookers. The wash-down type rice cooker produces soggy, mushy rice with low palatability for Koreans. Therefore, it was judged that low-carbohydrate rice cooked with a siphon-type rice cooker was more appropriate to replace regular rice. The cooking characteristics of these two low carbohydrate rice cookers are shown in the introduction section (Line 64-80).
- The authors did not disclose the composition of mixed-grain rice.
|
|
→ We used commercially available "11 kinds of mixed grains" for the study, and the following content was added to the method section of the paper.
|
- Rice was washed three times with water before placing in the cooker. What was the loss of macro-and micronutrients by the washing?
→Washing rice with water three times before cooking has already been used in many studies as a standardized rice washing method. Unfortunately, there is no study examining the difference in micronutrient content between rice cooked with unwashed rice and rice cooked with rice washed three times. The purpose of this study is to evaluate the effects of low-carbohydrate rice intake and blood glucose changes, and it is considered that the effects of micronutrient changes that may occur during the washing process of rice on blood glucose are very low. We added three previous studies with a method of cooking rice by washing rice 3 times to the reference.
(Line 130) Rice [white rice (WR): polished rice 300 g; mixed-grain rice (MR): polished rice 210 g + mixed-grain 90 g] were washed three times with sufficient distilled water [18-20].
[18] Bhawamai S, Lin SH, Hou YY, Chen YH. Thermal cooking changes the profile of phenolic compounds, but does not attenuate the anti-inflammatory activities of black rice. Food Nutr Res. 2016, 60, e32941.
[19] Kim MK. Sensory Profile of Rice-Based Snack (Nuroongji) Prepared from Rice with Different Levels of Milling Degree. Foods. 2020, 9(6), 685.
[20] Kwak HS, Kim HG, Kim HS, et al. Sensory characteristics and consumer acceptance of frozen cooked rice by a rapid freezing process compared to homemade and aseptic packaged cooked rice. Prev Nutr Food Sci. 2013, 18(1), 67-75.
- Figure 3 (b): Is it the 'weight of water with the starch removed water' or the weight of rice with the starch removed water?
→ This is the weight of water transferred from the inner pot to the tray. This water contains starch dissolved from the rice, and the carbohydrate content of rice is reduced as much as the starch in the water removed through the tray in this way.
- Table 2. Protein content in low-carb MR appears significantly higher than others. What may be the reason? How many cooking replicates were taken to complete proximate analysis? Section 3.2 statistical analysis to find differences either within or between the groups are missing. they are important to find strength in the study and analysis.
→The protein content of regular rice and low-carb rice may seem to be numerically different, but the differences were not statistically significant. We added statistical analysis results in Table 2 (Page 7). Low-carb rice had significantly lower energy, carbohydrate content and water content than regular rice, and there was no difference in other nutritional components.
→ All were carried out in triplicate and repeated 3 times. This was added to Table2.
→ We performed statistical analysis on the test food composition results and presented the results in Table 2 (Page 7).
Table 2. Test food compositions
|
Test food |
Energy (kcal/100g) |
Carbohydrate (g/100g) |
Protein (g/100g) |
Fat (g/100g) |
Ash (g/100g) |
Dietary fiber (g/100g) |
Water (g/100g) |
|
Regular WR |
149.0 ± 1.01 |
34.0 ± 0.3 |
2.6 ± 0.0 |
0.4 ± 0.0 |
0.1 ± 0.0 |
0.4 ± 0.0 |
62.9 ± 0.3 |
|
Low-carb WR |
120.8 ± 3.7* |
27.6 ± 0.9* |
1.7 ± 0.8 |
0.5 ± 0.1 |
0.1 ± 0.0 |
0.4 ± 0.0 |
70.1 ± 0.8* |
|
Regular MR |
154.4 ± 2.3 |
34.8 ± 0.4 |
2.9 ± 0.1 |
0.5 ± 0.1 |
0.2 ± 0.0 |
0.5 ± 0.0 |
61.6 ± 0.4 |
|
Low-carb MR |
129.3 ± 2.9** |
28.0 ± 1.4** |
3.3 ± 1.0 |
0.6 ± 0.2 |
0.2 ± 0.0 |
0.5 ± 0.0 |
68.0 ± 0.6** |
WR, white rice; MR, mixture of mixed grains to WR (WR/MR ratio is 7:3). All were carried out in triplicate and repeated 3 times.
1 The data represent the mean ± standard deviation (SD) calculated by the data of cooked rice.
*p<0.01 compared to regular WR; ** p<0.01 compared to regular MR, after independent t-test.
- Table 4. after 60, 90, and 120 mins, there appears no difference in observed blood glucose level. Although, the authors have not done statistical analysis. Then where does the title matches with the study results?
→ We did perform statistical analysis and listed in Table 4. The average blood glucose levels of the study subjects for each postprandial time period (0, 15, 30, 45, 60, 90, 120min) showed no statistically significant difference from the start of a meal to 120 minutes after a meal. (This is also expressed in the footnotes of Table 4)
→ In this study, it was in the iAUC that the difference in glycemic response was found when eating low-carb WR compared to that of regular WR. Blood glucose iAUC was analyzed as the average blood glucose for each time period and shown in Figure 4, and the hypoglycemic effect of low-carb WR was found in iAUC 0~15 min and 0~30 min postprandial.
→ Delaying the steep rise of blood glucose is beneficial for blood glucose management and can eventually lower postprandial blood glucose spikes resulting in stable blood glucose levels [31]. Therefore, these results show the positive effects of low-carb WR against postprandial spikes in blood glucose and can suggest options for those who hesitate to consume white rice due to the burden of blood glucose control.
- Discussion finds very poor support from relevant references and also convincing arguments.
→There is a limit to reviewing the results of relevant references as the concept of low carb rice cooker is an emerging interest and has not yet been studied thoroughly. However, the purpose of this study is to give people some scientific-evidence of using the rice cooker rather than to give an academic explanation. We think that this paper lays an important foundation at the present time when the demand and supply of low-carb rice cookers are rising. Thanks to the reviewers' comments, we tried to enrich the discussion section by additionally reviewing 4 related papers.
(Line 318) Low-carbohydrate diets, which has recently attracted a lot of attention, are known to be effective in weight loss and prevention of chronic diseases such as diabetes by regulating the glucose-insulin axis [26]. In women with gestational diabetes or overweight or obese adults, low-carbohydrate mixed meals had significantly reduced iAUC in blood glucose and/or insulin [26,27]. In addition, it was also reported that intake of a low-carbohydrate ketogenic diet for 12 weeks had an effect of reducing body weight and BMI [28]. Also, there is the gene expression involved in nutrient metabolism is different during day and night and depends on the meal composition [29]. Therefore, manipulating the meal type at night (e.g. more protein and less carbohydrate) may mitigate regulation of glucose uptake and fatty acid synthesis and oxidation [29]. Previous studies mainly evaluated the effect of a 'low-carbohydrate diet' with increased protein and fat content. Meanwhile, this study was to evaluate the blood glucose response in the low-carb rice intake. The effect of reducing blood glucose levels of low-carb rice can be suggested that the blood glucose can be controlled, not a high protein or high fat meal, through daily meals.

Reviewer 2 Report
Comments to Authors
This study showed that possible health benefits of low-carb WR in reducing early postprandial spikes in the blood glucose level without significant differences in the appetite.
Authors are kindly requested to emphasize the current concepts about these issues in the context of recent knowledge and the available literature. This articles should be quoted in the References list.
References
- Capillary Triglycerides in Late Pregnancy-Challenging to Measure, Hard to Interpret: A Cohort Study of Practicality. Nutrients. 2021; 13 (4): 1266. Published 2021 Apr 13. doi:10.3390/nu13041266-
- Diurnal variation in gene expression of human peripheral blood mononuclear cells after eating a standard meal compared with a high protein meal: A cross-over study. Clin Nutr. 2021; 40 (6): 4349-4359. doi:10.1016/j.clnu.2021.01.011.
- Reducing the glycemic index or carbohydrate content of mixed meals reduces postprandial glycemia and insulinemia over the entire day but does not affect satiety. Diabetes Care. 2012; 35 (8): 1633-1637. doi:10.2337/dc12-0329.
- Effects of a Low-Carbohydrate Ketogenic Diet on Reported Pain, Blood Biomarkers and Quality of Life in Patients with Chronic Pain: A Pilot Randomized Clinical Trial. Pain Med. 2022; 23 (2): 326-338. doi:10.1093/pm/pnab278.
Author Response
The revised manuscript was sent by e-mail (foods@mdpi.com).
Thank you for review's comments and opportunities to improve our manuscript ‘Postprandial glucose response after consuming low-carbohydrate, low-calorie rice cooked in a carbohydrate-reducing rice cooker’
We thank the reviewers for their detailed and specific comments for the manuscript to be much improved.
Sincerely yours,
-------------------------------------------------------------------------------
This study showed that possible health benefits of low-carb WR in reducing early postprandial spikes in the blood glucose level without significant differences in the appetite.
Authors are kindly requested to emphasize the current concepts about these issues in the context of recent knowledge and the available literature. This article should be quoted in the References list.
References
- Capillary Triglycerides in Late Pregnancy-Challenging to Measure, Hard to Interpret: A Cohort Study of Practicality. Nutrients. 2021; 13 (4): 1266. Published 2021 Apr 13. doi:10.3390/nu13041266-
- Diurnal variation in gene expression of human peripheral blood mononuclear cells after eating a standard meal compared with a high protein meal: A cross-over study. Clin Nutr. 2021; 40 (6): 4349-4359. doi:10.1016/j.clnu.2021.01.011.
- Reducing the glycemic index or carbohydrate content of mixed meals reduces postprandial glycemia and insulinemia over the entire day but does not affect satiety. Diabetes Care. 2012; 35 (8): 1633-1637. doi:10.2337/dc12-0329.
- Effects of a Low-Carbohydrate Ketogenic Diet on Reported Pain, Blood Biomarkers and Quality of Life in Patients with Chronic Pain: A Pilot Randomized Clinical Trial. Pain Med. 2022; 23 (2): 326-338. doi:10.1093/pm/pnab278.
→ Thank you for suggesting papers for review. We improved the discussion section through the review of the papers.
(Line 138) Low-carbohydrate diets, which has recently attracted a lot of attention, are known to be effective in weight loss and prevention of chronic diseases such as diabetes by regulating the glucose-insulin axis [26]. In women with gestational diabetes or overweight or obese adults, low-carbohydrate mixed meals had significantly reduced iAUC in blood glucose and/or insulin [26,27]. In addition, it was also reported that intake of a low-carbohydrate ketogenic diet for 12 weeks had an effect of reducing body weight and BMI [28]. Also, there is the gene expression involved in nutrient metabolism is different during day and night and depends on the meal composition [29]. Therefore, manipulating the meal type at night (e.g. more protein and less carbohydrate) may mitigate regulation of glucose uptake and fatty acid synthesis and oxidation [29]. Previous studies mainly evaluated the effect of a 'low-carbohydrate diet' with increased protein and fat content. Meanwhile, this study was to evaluate the blood glucose response in the low-carb rice intake. The effect of reducing blood glucose levels of low-carb rice can be suggested that the blood glucose can be controlled, not a high protein or high fat meal, through daily meals.

Reviewer 3 Report
Comments to the Author
This is a practically important article indicates that Postprandial glucose response after consuming low-carbohydrate, low-calorie rice cooked in a carbohydrate-reducing rice cooker.
However, it is necessary to major revision the research method etc. in several respects.
- Introduction
Carb-reducing electric rice cookers theoretically reduce the starch contents in rice.
Does this method decrease the amount of protein, vitamins, minerals, etc.?
- Materials and Methods
Are subjects with impaired glucose tolerance included in this study?
- Materials and Methods
Sample size determination was based on the method of the International Organization for Standardization. This suggests that using 10 subjects to measure the glycemic response to a single food could exhibit adequate power and accuracy.
Please add the following information.
Effect size, Power
- Results
Please add the following information.
Minimum and maximum value of fasting blood glucose.
- Discussion
From the results of the study, iAUC at 15 and 30 minutes after the consumption of low-carb WR was significantly lower than that of regular WR. Even a small reduction of blood glucose response 15–30 minutes after a meal can reportedly have a beneficial effect on blood glucose management in patients with diabetes. Therefore, these results show the positive effects of low-carb WR against postprandial spikes in blood glucose.
Please add references to the following discussion.
Even a small reduction of blood glucose response 15–30 minutes after a meal can reportedly have a beneficial effect on blood glucose management in patients with diabetes.
This study failed to reveal the glycemic efficacy of a low-carb diet at 120 minutes postprandial.
Please discuss the results of this results.
- The strengths of the study
Third, the discovery that there was no difference in the feeling of fullness, hunger, and DTE between consumption of low-calorie low-carb WR and regular rice, we found the possibility that low-carb WR could affect body weight loss.
In this study, cannot mention body weight loss. Please revise the following sentence.
Third, the discovery that there was no difference in the feeling of fullness, hunger, and DTE between consumption of low-calorie low-carb WR and regular rice.
Author Response
The revised manuscript was sent by e-mail (foods@mdpi.com).
Thank you for review's comments and opportunities to improve our manuscript ‘Postprandial glucose response after consuming low-carbohydrate, low-calorie rice cooked in a carbohydrate-reducing rice cooker’
We thank the reviewers for their detailed and specific comments for the manuscript to be much improved.
Sincerely yours,
-------------------------------------------------------------------------------
This is a practically important article indicates that Postprandial glucose response after consuming low-carbohydrate, low-calorie rice cooked in a carbohydrate-reducing rice cooker. However, it is necessary to major revision the research method etc. in several respects.
- Introduction: Carb-reducing electric rice cookers theoretically reduce the starch contents in rice. Does this method decrease the amount of protein, vitamins, minerals, etc.?
→ The protein content of regular rice and low-carb rice may seem to be numerically different, but there was no statistical significant difference. We added statistical analysis results in Table 2. Low-carb rice had significantly lower energy, carbohydrate content and water content than regular rice, and there was no difference in other nutritional components including protein. Unfortunately, vitamins and minerals were not analyzed. There is a possibility of loss of nutrients other than carbohydrates due to the principle of low-carb rice cooker that cooks rice with excess water, but we think that the main purpose of low-carb rice is to control blood glucose. This study is significant because it is the first study to evaluate the blood glucose response of low-carb rice cooked in a low-carb rice cooker, and more in-depth analysis should be performed through additional studies. We performed statistical analysis on the test food composition results and presented the results in Table 2.
Table 2. Test food compositions
|
Test food |
Energy (kcal/100g) |
Carbohydrate (g/100g) |
Protein (g/100g) |
Fat (g/100g) |
Ash (g/100g) |
Dietary fiber (g/100g) |
Water (g/100g) |
|
Regular WR |
149.0 ± 1.01 |
34.0 ± 0.3 |
2.6 ± 0.0 |
0.4 ± 0.0 |
0.1 ± 0.0 |
0.4 ± 0.0 |
62.9 ± 0.3 |
|
Low-carb WR |
120.8 ± 3.7* |
27.6 ± 0.9* |
1.7 ± 0.8 |
0.5 ± 0.1 |
0.1 ± 0.0 |
0.4 ± 0.0 |
70.1 ± 0.8* |
|
Regular MR |
154.4 ± 2.3 |
34.8 ± 0.4 |
2.9 ± 0.1 |
0.5 ± 0.1 |
0.2 ± 0.0 |
0.5 ± 0.0 |
61.6 ± 0.4 |
|
Low-carb MR |
129.3 ± 2.9** |
28.0 ± 1.4** |
3.3 ± 1.0 |
0.6 ± 0.2 |
0.2 ± 0.0 |
0.5 ± 0.0 |
68.0 ± 0.6** |
WR, white rice; MR, mixture of mixed grains to WR (WR/MR ratio is 7:3) ). All were carried out in triplicate and repeated 3 times.
1 The data represent the mean ± standard deviation (SD) calculated by the data of cooked rice.
*p<0.01 compared to regular WR; ** p<0.01 compared to regular MR, after independent t-test.
- Materials and Methods: Are subjects with impaired glucose tolerance included in this study?
→ No participants were diagnosed with impaired glucose tolerance. However, as the result of OGTT, there were two patients whose postprandial blood glucose level was 140-199 mg/dL. Since the effect of low-carb rice on blood glucose response may be more effective for those who need blood glucose control, we included only those who could not control blood glucose even with drugs in the exclusion criteria of the study participants.
- Materials and Methods: Sample size determination was based on the method of the International Organization for Standardization. This suggests that using 10 subjects to measure the glycemic response to a single food could exhibit adequate power and accuracy. Please add the following information. Effect size, Power
→ Unfortunately, the effect size and power of ISO 26642 could not be found. However, this method is still recognized for its validity as a standard cited in many studies. We have added to our reference a recent study citing this study.
[22] Dall'Asta, M., Dodi, R., Di Pede, G., Marchini, M., Spaggiari, M., Gallo, A., ... & Scazzina, F. Postprandial blood glucose and insulin responses to breads formulated with different wheat evolutionary populations (Triticum aestivum L.): A randomized controlled trial on healthy subjects. Nutrition, 2022, 94,111533.
- Results: Minimum and maximum value of fasting blood glucose.
→ This has been added to Table 1.
Table 1. Characteristic of the subjects
|
Classification |
Subjects (n = 13) |
|
Age (years) |
56.8 ± 12.1 |
|
Sex (males/females) |
4/9 |
|
Height (cm) |
162.2 ± 10.6 |
|
Body weight (kg) |
64.7 ± 14.7 |
|
Body mass index (kg/m²) |
24.5 ± 4.4 |
|
% Body fat (%) |
32.4 ± 5.3 |
|
Skeletal muscle mass (kg) |
24.0 ± 6.9 |
|
Waist-Hip Ratio |
0.90 ± 0.05 |
|
Fasting blood glucose (mg/dL, range) |
99.5 ± 15.0 (82-120) |
1 Data are mean ± standard deviation (SD)
- Discussion: From the results of the study, iAUC at 15 and 30 minutes after the consumption of low-carb WR was significantly lower than that of regular WR. Even a small reduction of blood glucose response 15–30 minutes after a meal can reportedly have a beneficial effect on blood glucose management in patients with diabetes. Therefore, these results show the positive effects of low-carb WR against postprandial spikes in blood glucose. Please add references to the following discussion.
Even a small reduction of blood glucose response 15–30 minutes after a meal can reportedly have a beneficial effect on blood glucose management in patients with diabetes.
→ This sentence means that slowing the rise in blood glucose can be beneficial for blood glucose management and can help with blood glucose spikes. We changed some sentences and added references.
(Line 341) Delaying the rise of blood glucose is beneficial for blood glucose management and can help lower postprandial blood glucose spikes resulting in stable blood glucose levels [31].
[31] Jenkins DJ, Kendall CW, Augustin LS, Franceschi S, Hamidi M, Marchie A, Jenkins AL, Axelsen M. Glycemic index: overview of implications in health and disease. Am J Clin Nutr. 2002, 76(1), 266S-73S.
- This study failed to reveal the glycemic efficacy of a low-carb diet at 120 minutes postprandial. Please discuss the results of this results.
→ As the body secretes the hormone insulin, blood glucose usually returns to the water before a meal 2 hours after a meal. Even in various previous studies evaluating the effect of diet or food intake on the blood glucose response, it is difficult to find a significant difference in blood glucose between the control group and the intervention group at 2 hours after a meal, even if there is a difference in the blood glucose response up to about 60 minutes after a meal.
Ballance, S., Knutsen, S. H., Fosvold, Ø. W., Fernandez, A. S., & Monro, J. Predicting mixed-meal measured glycaemic index in healthy subjects. European journal of nutrition, 2019, 58(7), 2657-2667.
Boravek, D., Duncan, A. M., VanderSluis, L. B., Turkstra, S. J., Rogers, E. J., Wilson, J. M., ... & Ramdath, D. D. Carbohydrate replacement of rice or potato with lentils reduces the postprandial glycemic response in healthy adults in an acute, randomized, crossover trial. The Journal of nutrition, 2018, 148(4), 535-541.
Zhu, R., Fan, Z., Li, G., Wu, Y., Zhao, W., Ye, T., & Wang, L. A comparison between whole grain and pearled oats: acute postprandial glycaemic responses and in vitro carbohydrate digestion in healthy subjects. European Journal of Nutrition, 2020, 59(6), 2345-2355.
Da Costa, T. H., Reis, C. E., da Silva, F. V., & Casulari, L. A. Improvement in metabolic parameters in obese subjects after 16 weeks on a Brazilian-staple calorie-restricted diet. Nutrition Research and Practice, 2014, 8(4), 410-416.
Chiavaroli, L., Di Pede, G., Dall’Asta, M., Cossu, M., Francinelli, V., Goldoni, M., ... & Brighenti, F. The importance of glycemic index on post-prandial glycaemia in the context of mixed meals: A randomized controlled trial on pasta and rice. Nutrition, Metabolism and Cardiovascular Diseases, 2021, 31(2), 615-625.
→ Therefore, it is thought that the fact that there was no difference in blood glucose 2 hours after a meal between the regular rice and low-carb rice intake groups is a natural phenomenon due to the insulin hormone response. We think that this study is meaningful in that it is the first paper to evaluate the blood glucose response of low-carb rice and that it can give options to those who need blood glucose control among those who eat rice as their main food. And it is expected that the high demand and interest in low-carb rice cookers will lead many studies of low-carb rice using this study as a cornerstone. A sentence was added to the review from this point of view.
(Line 343) Therefore, these results show the positive effects of low-carb WR against postprandial spikes in blood glucose and can suggest options for those who hesitate to consume white rice due to the burden of blood glucose control.
- The strengths of the study: Third, the discovery that there was no difference in the feeling of fullness, hunger, and DTE between consumption of low-calorie low-carb WR and regular rice, we found the possibility that low-carb WR could affect body weight loss. In this study, cannot mention body weight loss. Please revise the following sentence.
→This has changed.
(Line 417) Third, the discovery that there was no difference in the feeling of fullness, hunger, and DTE between consumption of low-calorie low-carb WR and regular rice.

Round 2
Reviewer 1 Report
The authors have satisfactorily responded to the issues raised in the manuscript. They have also included appropriate corrections in the manuscript.
As this research has a different focus, it is obvious authors may not find much literature support for the same.
I my opinion, the manuscript may be accepted for publication.
Reviewer 3 Report
No comment